# Factors associated with premarital HIV testing among married women in Ethiopia

**Mohammed Ahmed**[1]*, **Abdu Seid**[2]

**1** Department of Public Health, College of Health Science, Woldia University, Woldia, Ethiopia, **2** Department of Midwifery, College of Health Science, Woldia University, Woldia, Ethiopia

* mohaasrar12@gmail.com

## Abstract

### Background

Premarital HIV testing is the key entry point in prevention, care, treatment, and support services, in which people learn their HIV status and its implications to make informed decisions about their health. This study was, therefore, conducted to identify factors associated with premarital HIV testing among married women in Ethiopia.

### Methods

A cross-sectional study design was used, and secondary data analysis was done using 2016 Ethiopian demographic health survey (EDHS). Two-stage stratified cluster sampling technique was used. The data were analyzed by using SPSS version 20. Frequencies and weighted percentage of the variables, and second-order Rao-Scott statistic were computed. Multivariate logistic regression analysis was performed to control confounders and to identify predictors of premarital HIV testing. Adjusted odds ratio with 95% confidence interval was considered to declare statistically significant associations.

### Result

The total sample comprised 9602 married women. In this study, the odds of premarital HIV testing were associated with being urban residents (**AOR: 1.81; 95% CI: 2.74–5.20**), attended primary education (AOR:**1.54; 95%:1.27–1.87**), secondary education **(AOR:2.34; 95% CI:1.70–3.23),** higher education **(AOR:**2.92**; 95% CI:1.90–4.50),** access to media **(AOR: 1.44; 95% CI:1.20–1.76),** being rich **(AOR: 1.52; 95%CI:1.12–2.07),** andrichest **(AOR: 1.67;95%CI:1.15–2.44),** known the place of HIV testing **(AOR: 4.95; 95% CI:3.44–7.11),** discriminatory attitude to PLHIV **(AOR: 1.47; 95%CI:1.23–1.76),** being khat chewer **(AOR:** 1.60**;95%CI:1.11–2.31**), and alcohol drinker **(AOR: 1.55; 95% CI:1.27–1.90)**.

### Conclusion

It is possible to conclude that being urban resident, attending education (primary, secondary, higher), media access, improved wealth index, knowing the places for HIV testing, chewing khat, drinking alcohol, and having discriminatory attitude towards PLHIV were positively associated with premarital HIV testing. The Ethiopian government needs to step up

**Data Availability Statement:** For this analysis, we used the USAID–DHS program 2016 Ethiopian demographic and health survey data set. To request the same or different data for another purpose, a new research project request should be

submitted to the DHS program here: https://
dhsprogram.com/data/Access-Instructions.cfm.
The DHS Program will normally review all data
requests within 24 – 48 hours (during working
days) and provide notification if access has been
granted, or if additional project information is
needed before access can be granted. After
receiving permission, the researcher can login and
select the specific data in the format they prefer.

**Funding:** The author(s) received no specific
funding for this work

**Competing interests:** The authors have declared
that no competing interests exist.

efforts to expand education for all Women. Advancing access to HIV testing for rural women
may also increase premarital HIV testing services uptake. Further qualitative researches
need to be done to assess the relationship between discriminatory attitude towards PLHIV
and premarital HIV testing.

## Introduction

Globally, Human Immune Virus/Acquired Immune Deficiency Syndrome (HIV/AIDS) epi-
demic stage has been expanding at different times due to different risk factors that curb down
the general productivity of the community, and the national economy [1].

Evidence showed that 71% of people living with HIV/AIDS found in Sub-Saharan Africa;
among these women are more affected than men [2]. Ethiopia is one of the highly affected
countries with HIV/AIDS pandemic as early as 1985 and the disease has spread at alarming
rate throughout the country [3]. In Ethiopia in 2018 report, the prevalence of HIV among
adults (15–49 years) was 1%, from this figure women account 63.08% [4].

A study done in New Jersey, United states of America (USA) showed that seroprevalence in
unmarried couples were 0.55–0.62%. For this reason, voluntary HIV counseling and testing for
marriage applicants is recommended [5]. As well in Africa, voluntary HIV counseling and testing
has been acknowledged as cost-effective measure for the prevention and control of HIV [6, 7].

The government of Ethiopia has acceptedpremarital voluntary HIV counseling and testing,
which is recommended by World Health Organizations (WHO), since it is one of the key ele-
ments in the prevention and control of HIV/AIDS in the country [8, 9]. Premarital voluntary
HIV counseling and testing offers an opportunity where prospective couples can know their
HIV status before marriage [8]. Therefore, it is one of the renowned strategies on preventing
both heterosexual and vertical transmission of HIV [10–12].

According to Ethiopia demographic and health Survey (EDHS) 2016 report, the proportion
of women and men who were ever tested for HIV increased from 2% for women and men in
2005 to 20% for women and 21% for men in 2011. However, the HIV testing coverage remains
unchanged between 2011 and 2016. Furthermore, 24.5% of married women aged 15–49 ever
tested before getting married or living with a partner [13].

Undeniably, women are particularly vulnerable to HIV infection because of increased bio-
logic susceptibility to HIV transmission through heterosexual contact [14, 15]. Women are
also at increased risk of HIV because they face a host of structural barriers and contextual gen-
der inequalities such as poverty, economic disempowerment, cultural inequities, increased risk
of sexual violence, and gender power imbalance in sexual interactions [16, 17].

A clinical trial study showed that about 65–85% of new infections are acquired from the
married/cohabiting partner was due to HIV sero-discordant couples [18]. HIV positive indi-
viduals in sero-discordant marriages results in increased risk for HIV negative partners [19].
In the same vein, a study conducted in Zambia and Rwanda showed that an estimated 50% of
new heterosexual HIV infections crop up among sero-discordant couples [20].

Various studies showed that being urban resident [21], having least stigmatizing attitudes [22,
23], attending secondary and higher education [22], women with aged 25–34 years and 35+ [21],
being rich in wealth index, divorced/widowed [24], undergone sexual intercourse [24], drinking
alcohol [25] were positively associated with premarital HIV testing. On the other hand, unable to
read and write [24], being unemployed [26], were inversely related to premarital HIV testing.

Despite the aforementioned factors, different studies reveal inconsistent results; for
instance, being older aged [26], urban resident [23], having comprehensive knowledge about

HIV/AIDS [21], knowing the places for HIV testing [21], were less likely to be tested for HIV. However, a study done in Northeast Ethiopia showed that having comprehensive knowledge about HIV is risk factor for premarital HIV testing [24].

Step up prevention activities require a thorough understanding of the HIV epidemic character, modes of transmission and populations affected as these inform the extent to which evidence based modalities can be adapted and pooled to substantially reduce HIV transmission, which is critical in continuing the path to avert epidemic trajectory [27, 28].

In a nutshell, premarital testing is the best solution for prevention, care, treatment, and support services [29]. Make out and intervening thus factors enable for undertakes premarital HIV testing, which leads to trim down HIV acquisition among couples. Therefore, the endeavor of this study was to identify factors associated with premarital HIV testing among married women in Ethiopia.

## Methods and materials

### Data

The current study uses secondary data from 2016 Ethiopia Demographic and Health Survey (EDHS). A detailed description of the study design and methodology of the 2016 is found elsewhere (13). A nationally representative sample was obtained based on a two-stage cluster sampling. The first and second stages involved the selection of the clusters and households in each cluster, respectively. Further, stratification by rural-urban areas was taken into account. This study was based on data from the Woman's Questionnaire, which was administered to all women aged 15–49 in the selected households. The analytic sample for the current study consisted of married women aged 15–49 years (**n = 9602**).

### Outcome of interest: Premarital HIV testing

The main outcome of interest was self-reported history of premarital HIV testing among married women (yes/no). The independent variables were selected based on literature review which deemed to be the factors associated with premarital HIV testing and includes age, education status, type of residence, occupation, wealth index, media access, knowing the places for HIV testing, comprehensive knowledge about HIV, discriminatory attitude to PLHIV, khat chewing, and alcohol drinking.

*Comprehensive knowledge about HIV/AIDS* was defined based on a widely used measure where each woman was asked whether or not she agreed or disagreed with the following five items: (1) Consistent use of condoms during sexual intercourse can reduce the chance of getting HIV; (2) having just one uninfected faithful partner can reduce the chance of getting HIV; (3) Healthy-looking person can have HIV; (4) HIV can be transmitted by mosquito bites; and (5) a person can become infected by sharing food with a person who has HIV. An additive summary score was created and which was then dichotomized to create a binary variable with 0 indicating at least one incorrect response and 1 to indicate correct response to five items.

*Media exposure* was defined based on response to how often respondents read a newspaper, listened to the radio, or watched television. Those who responded at least once a week to any of these sources were considered to have access to media/media exposure.

*Discriminatory attitudes towards people living with HIV (PLHIV)* was defined based on the response to two items: (1)Would not buy fresh vegetables from a shopkeeper or vendor if they knew that person had HIV (yes/no); (2)Children living with HIV should not be allowed to attend school with children who do not have HIV(yes/no). Respondents having discriminatory attitudes towards PLHIV are those who responded yes for the above questions otherwise not.

## Statistical analysis

The data were analyzed by using Statistical Package for Social Science (SPSS) version 20. All statistical procedures incorporated complex sampling design analysis applied in the 2016 EDHS. Frequencies and weighted percentage of study variables were calculated. Rao–Scott chi-square test was used to examine the relationship between premarital HIV testing and each of the independent variables. Multivariate logistic regression analysis was performed to control confounders and to identify independent predictors about premarital HIV testing. All independent variables were entered in the multivariate logistic regression model irrespective of the p-values in the bivariate analysis. Adjusted odds ratio (AOR) with 95% confidence interval was used to declare statistically significant associations.

## Ethics approval and consent to participate

Ethical clearance for the study is not required since it is secondary data analysis from EDHS 2016 data base. The researchers have received the survey data from USAID–DHS program and then the researchers of this study have maintained the confidentiality of the data.

## Results

### Participant's characteristics

A total of 9602 sub-sample of married women within the EDHS 2016 were included and analyzed. Majority of respondents (84.2%) were rural residents and 23.5% were found in the age range between 25–29 years. Regarding educational status, 61.8% of the respondents didn't attend formal education. Only, 96.8% of married women's didn't have comprehensive knowledge about HIV, and 62.2% didn't have access to media (**Table 1**).

### Factors associated with premarital HIV testing among married women in Ethiopia

All the variables were entered intomultivariate logistic regression analysis. After adjusting for potential confounders by logistic regression, being rural resident, education attainment (primary, secondary, higher), media access, being rich and richest, knowing the places for HIV testing, chewing khat, drinking alcohol, and having a discriminatory attitude towards PLHIV were positively associated with premarital HIV testing.

In this study, the odds of premarital HIV testing were **1.81**[**AOR: 1.81 (1.31–2.50)**] times higher among urban compared to rural residents. Likewise, the odds of premarital HIV testing among women who attended primary, secondary, and higher education was 1.54[AOR:**1.54 (1.27–1.87)**], 2.34[AOR:**2.34(1.70–3.23)**], **2.92**[**AOR:2.92(1.90–4.50)** timeshigher compared to those who did not attend formal education respectively. In addition, the odds of premarital HIV testing were **1.44** times [AOR: **1.44(1.20–1.76)**] higher among women who had media access than its counter parts. The odds of premarital HIV testing were **1.52**[**AOR: 1.52 (1.12–2.07)**]**, and 1.67**[**AOR: 1.67(1.15–2.44)**]times higher among women who had richer and richest wealth index category, compared to the poorest respectively.

The odds of premarital HIV testing were **4.95**[**AOR: 4.95 (3.44–7.11)**] times higher among participants who had known the places for HIV testing compared to its counter parts.

The study revealed that the odds of premarital HIV testing were **1.47**[**AOR: 1.47 (1.23–1.76)**] times higher among participants who had a discriminatory attitude to PLHIV than its counterparts. Moreover, the odds of premarital HIV testing were **1.60**[**AOR: 1.60 (1.11–2.31)**]**, and 1.55**[**AOR: 1.55 (1.27–1.90)**] times higher among khat chewer and alcohol drinker compared to their counterparts respectively (**Table 2**).

**Table 1. Characteristics of respondents by premarital HIV testing (n = 9602).**

| Variables | Category | Overall | Premarital HIV testing | | p-value |
|---|---|---|---|---|---|
| | | | No | Yes | |
| | | n(wt.%) | n (wt. %) | n (wt. %) | |
| Residence | Urban | 2369(15.8) | 1109 (9.1) | 1260(37.1) | p<0.001 |
| | Rural | 7233(84.2) | 6033(90.9) | 1200(62.9) | |
| Age | 15–19 | 641(5.7) | 433(5.1) | 208(7.5) | p<0.001 |
| | 20–24 | 1725(16.5) | 1091(14.0) | 634(24.6) | |
| | 25–29 | 2189(23.5) | 1470(21.1) | 719(31.1) | |
| | 30–34 | 1814(20.1) | 1355(20.5) | 459(19) | |
| | 35–39 | 1536(15.8) | 1289(17.7) | 247(9.8) | |
| | 40–44 | 1006(10.5) | 889(12.4) | 117(4.6) | |
| | 45–49 | 691(7.8) | 615(9.2) | 76 (3.4) | |
| Educational status | No education | 5625(61.8) | 4948(70.4) | 677(34.8) | p<0.001 |
| | Primary | 2621(28.1) | 1708(25.0) | 916(37.8) | |
| | Secondary | 839(6.2) | 338(3.1) | 501(15.8) | |
| | Higher | 517(3.9) | 151(1.5) | 366(11.5) | |
| Occupation | Unemployed | 5273(52.0) | 4117(53.7) | 1156(46.7) | p<0.001 |
| | Agricultural | 1890(23.5) | 1580(25.4) | 310(17.6) | |
| | Non-agricultural | 2439(24.5) | 1445(20.9) | 994(35.7) | |
| Access to media | No | 5799(62.2) | 5025(69.6) | 774(38.6) | p<0.001 |
| | Yes | 3803(37.8) | 2117(30.4) | 1686(61.4) | |
| Wealth index | Poorest | 2880(19.2) | 2598(22.3) | 282(9.3) | p<0.001 |
| | Poorer | 1474(20.4) | 1248(22.8) | 226(12.7) | |
| | Middle | 1342(20.3) | 1083(21.7) | 259(18.7) | |
| | Richer | 1289(19.6) | 983(19.6) | 306(19.6) | |
| | Richest | 2617(20.5) | 1230(13.5) | 1387(42.5) | |
| Comprehensive knowledge about HIV | No | 9396(96.8) | 6974(96.5) | 2422(97.5) | 0.132 |
| | Yes | 206(3.2) | 168 (3.5) | 38 (2.5) | |
| Know the places to HIV testing | No | 1872(25.7) | 1789(32.6) | 83 (5.7) | p<0.001 |
| | Yes | 6810(74.3) | 4470(67.4) | 2340 (94.3) | |
| Discriminatory attitude | No | 6521(74.2) | 5463(81.1) | 1058 (52.4) | p<0.001 |
| | Yes | 3081(25.8) | 1679(18.9) | 1402 (47.6) | |
| Chewing khat | No | 8493(85.2) | 6336(84.8) | 2156(86.6) | 0.406 |
| | Yes | 1109(14.8) | 805(15.2) | 304(13.4) | |
| Alcohol drinking | No | 6651(65.4) | 5232(68.2) | 1419(56.9) | p<0.001 |
| | Yes | 2951(34.6) | 1910(31.8) | 1041(43.1) | |

## Discussion

Premarital voluntary HIV counseling and testing is one of the well-known strategies for preventing both heterosexual and vertical transmission of HIV. This study aimed at identifying factors associated with premarital HIV testing among married women in Ethiopia.

In this study, premarital HIV testing was positively associated with residence, educational status, media access, wealth index, knowing the place for HIV testing, chewing khat, drinking alcohol, and having discriminatory attitude towards PLHIV.

Considering residence, women who were residing in urban area have higher odds to undertake premarital HIV testing. This finding is consistent with a study done in Malawi [21], and Nigeria [30, 31]. The reason for this may be better availability and accessibility of HIV testing facilities in urban settings.

**Table 2. Multivariate analysis table for identifying factors associated with premarital HIV testing among married women in Ethiopia (n = 9602).**

| Variables | Category | Premarital HIV testing | | COR(95%CI) | AOR(95%CI) |
|---|---|---|---|---|---|
| | | No | Yes | | |
| | | n (wt. %) | n (wt. %) | | |
| Residence | Urban | 1109 (9.1) | 1260 (37.1) | 5.89(4.72–7.35) | **1.81(1.31–2.50)*** |
| | Rural | 6033 (90.9) | 1200 (62.9) | Ref | Ref |
| Age | 15–19 | 433 (5.1) | 208 (7.5) | Ref | Ref |
| | 20–24 | 1091 (14.0) | 634 (24.6) | 1.19(0.91–1.56) | 0.87(0.63–1.20) |
| | 25–29 | 1470 (21.1) | 719 (31.1) | 1.00(0.76–1.31) | 0.74(0.53–1.03) |
| | 30–34 | 1355 (20.5) | 459 (19) | 0.63(0.47–0.83) | 0.48(0.32–0.70) |
| | 35–39 | 1289 (17.7) | 247 (9.8) | 0.37(0.27–0.52) | 0.26(0.18–0.39) |
| | 40–44 | 889 (12.4) | 117(4.6) | 0.25(0.17–0.36) | 0.17(0.10–0.28) |
| | 45–49 | 615 (9.2) | 76 (3.4) | 0.24(0.16–0.39) | 0.18(0.01–0.34) |
| Educational status | No education | 4948 (70.4) | 677(34.8) | Ref | Ref |
| | Primary | 1708 (25.0) | 916(37.8) | 3.06(2.58–3.63) | **1.54(1.27–1.87)*** |
| | Secondary | 338 (3.1) | 501(15.8) | 10.2(7.77–13.3) | **2.34(1.70–3.23)*** |
| | Higher | 151 (1.5) | 366(11.5) | 15.7(11.0–22.3) | **2.92(1.90–4.50)*** |
| Occupation | Unemployed | 4117 (53.7) | 1156(46.7) | Ref | Ref |
| | Agricultural | 1580(25.4) | 310(17.6) | 0.79(0.65–0.98) | 0.98(0.77–1.22) |
| | Non-agricultural | 1445(20.9) | 994(35.7) | 1.96(1.59–2.41) | 0.97(0.79–1.19) |
| Access to media | No | 5025(69.6) | 774(38.6) | Ref | Ref |
| | Yes | 2117(30.4) | 1686(61.4) | 3.64(3.06–4.32) | **1.44(1.20–1.76)*** |
| Wealth index | Poorest | 2598(22.3) | 282(9.3) | Ref | Ref |
| | Poorer | 1248(22.8) | 226(12.7) | 1.34(1.02–1.76) | 1.05(0.77–1.42) |
| | Middle | 1083(21.7) | 259(18.7) | 1.74(1.32–2.30) | 1.26(0.94–1.71) |
| | Richer | 983(19.6) | 306(19.6) | 2.40(1.85–3.13) | **1.52(1.12–2.07)*** |
| | Richest | 1230(13.5) | 1387(42.5) | 7.54(5.71–9.94) | **1.67(1.15–2.44)*** |
| Comprehensive knowledge about HIV | No | 6974(96.5) | 2422(97.5) | Ref | Ref |
| | Yes | 168 (3.5) | 38 (2.5) | 0.70(0.44–1.11) | 1.27(0.80–2.02) |
| Know place to HIV testing | No | 1789(32.6) | 83 (5.7) | Ref | Ref |
| | Yes | 4470(67.4) | 2340 (94.3) | 7.94(5.58–11.3) | **4.95 (3.44–7.11)*** |
| Discriminatory attitude | No | 5463(81.1) | 1058 (52.4) | Ref | Ref |
| | Yes | 1679(18.9) | 1402 (47.6) | 3.90(3.29–4.61) | **1.47 (1.23–1.76)*** |
| Chewing khat | No | 6336(84.8) | 2156(86.6) | Ref | Ref |
| | Yes | 805(15.2) | 304(13.4) | 0.86(0.61–1.22) | **1.60 (1.11–2.31)*** |
| Alcohol drinking | No | 5232(68.2) | 1419(56.9) | Ref | Ref |
| | Yes | 1910(31.8) | 1041(43.1) | 1.62(1.34–1.96) | **1.55 (1.27–1.90)*** |

Ref- reference category AOR: Adjusted Odds ratio; COR: Crude Odds Ratio; * P-value < 0.05

The study further showed that women who were educated have higher odds to carry out premarital HIV testing. This finding is in line with a study conducted in Kenya [32] and Uganda [33]. This could be elucidated by educated women take care of HIV infection, as they easily understood both the transmission and prevention methods [33].

Regarding media access and wealth index, women who had media access and were richer and richest have higher odds to undertake premarital HIV testing. This could be expounded by the possibility that higher income for women enhances their status in the household, enables to be educated and can help to have better access to media easily without constraints [33].

Concerning the use of alcohol and khat, the proportion of women who consumed alcohol and chewed chat in Ethiopia in the last 30 days was 50% and 65%, respectively [13]. As well, the present study revealed that premarital HIV testing was higher among khat chewers and alcohol drinkers compared to their counterparts. This could be due to risky sexual behavior after alcohol and khat chewing. This may have increased perceived susceptibility to HIV which in turn leads them to be tested for HIV [34, 35, 36].

The odds of premarital HIV testing were higher among women who knew the place for HIV testing. This finding is contrasts with a study conducted in Gambela region, which is found in Ethiopia [23]. This discrepancy may be due to sample size variation, in which the current study was done based on nationally representative data. Surprisingly, the study revealed that the odds of premarital HIV testing were higher among participants who had discriminatory attitude towards PLHIV than its counterparts. This finding is contrasts with the studies conducted in Uganda [22], South Africa [37], Nigeria [38] and Gambela region, which is found in Ethiopia [23].

The strength of this analysis is that it was based on nationally representative data with a large sample size. However, we cannot assign causations to any of the associations between the identified factors and the outcomes of interest due to cross sectional data.

## Conclusions

From the findings of the study, it is possible to conclude that being urban resident, attending education (primary, secondary, higher), having better media access, improved wealth index, knowing the places for HIV testing, chewing khat, drinking alcohol and having a discriminatory attitude towards PLHIV were positively associated with premarital HIV testing. The Ethiopian government needs to step up efforts to expand education for all women. Advancing access to HIV testing for rural women may also increase premarital HIV testing services uptake. Further qualitative researches need to be done to assess the relationship between discriminatory attitude towards PLHIV and premarital HIV testing.

## Supporting information

**S1 Checklist. STROBE statement—checklist of items that should be included in reports of *cross-sectional studies*.**
(DOC)

## Acknowledgments

We are grateful to the USAID–DHS program for providing access to the 2016 Ethiopian Demographic Health Survey

## Author Contributions

**Conceptualization:** Mohammed Ahmed, Abdu Seid.

**Data curation:** Mohammed Ahmed, Abdu Seid.

**Formal analysis:** Mohammed Ahmed.

**Methodology:** Mohammed Ahmed.

**Writing – original draft:** Mohammed Ahmed, Abdu Seid.

**Writing – review & editing:** Mohammed Ahmed, Abdu Seid.

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
