## [Decision Letter · Decision Letter 0]

15 Apr 2020

PONE-D-20-03022

Factors associated with premarital HIV testing among married women in Ethiopia.

PLOS ONE

Dear Mr Ahmed,

Thank you for submitting your manuscript to PLOS ONE. After careful consideration, we feel that it has merit but does not fully meet PLOS ONE’s publication criteria as it currently stands. Therefore, we invite you to submit a revised version of the manuscript that addresses the points raised during the review process.

We would appreciate receiving your revised manuscript by May 30 2020 11:59PM. To enhance the reproducibility of your results, we recommend that if applicable you deposit your laboratory protocols in protocols.io, where a protocol can be assigned its own identifier (DOI) such that it can be cited independently in the future. For instructions see: http://journals.plos.org/plosone/s/submission-guidelines#loc-laboratory-protocols

We look forward to receiving your revised manuscript.

Kind regards,

Benn Sartorius, PhD

Academic Editor

PLOS ONE

Journal Requirements:

1. We noticed you have some minor occurrence of overlapping text with the following previous publication(s), which needs to be addressed:

https://www.unaids.org/en/regionscountries/countries/ethiopia

https://www.benthamopen.com/FULLTEXT/TOAIDJ-10-34

In your revision ensure you cite all your sources (including your own works), and quote or rephrase any duplicated text outside the methods section. Further consideration is dependent on these concerns being addressed.

2. Please refrain from stating p values as 0.000, either report the exact value or employ the format p<0.001.

5. Your ethics statement must appear in the Methods section of your manuscript. If your ethics statement is written in any section besides the Methods, please move it to the Methods section and delete it from any other section. Please also ensure that your ethics statement is included in your manuscript, as the ethics section of your online submission will not be published alongside your manuscript.

Additional Editor Comments (if provided):

Please ensure that the revised manuscript is sent for a full professional English proofread.

Please include a completed STROBE checklist as part of the supplementary material for this article.

Reviewers' comments:

Reviewer's Responses to Questions

**Comments to the Author**

1. Is the manuscript technically sound, and do the data support the conclusions?

Reviewer #1: Yes

Reviewer #2: Yes

2. Has the statistical analysis been performed appropriately and rigorously? 

Reviewer #1: Yes

Reviewer #2: I Don't Know

3. Have the authors made all data underlying the findings in their manuscript fully available?

Reviewer #1: Yes

Reviewer #2: Yes

4. Is the manuscript presented in an intelligible fashion and written in standard English?

Reviewer #1: No

Reviewer #2: Yes

5. Review Comments to the Author

Reviewer #1: This is an important topic and the analysis is solid. The manuscript, however, needs extensive editing for English. It is not publishable in its current state.

Other concerns to address:

• Include the URL of where the original data can be accessed as per PLOS ONE guidelines

• Add background information about premarital HIV testing. Is it required? Is it encouraged? If so, by whom, how and to what extent? In general, what percent of women undergo premarital HIV testing, and what percent of men (to compare)? Is premarital HIV testing an important imperative of the Ministry of Health?

• Add information about gender-related factors in Ethiopia. While HIV prevalence is higher for women, what gender-related factors account for that? Are women more likely to be tested than men? Are women more at risk of contracting HIV due to gender-normative behavior?

• The Discussion section should be fleshed out more. For example, testing rates are higher among urban residents (it appears that part of your paragraph is missing). Could this be because they have better/closer/easier access to testing facilities? Is stigma lower in urban areas leading more women to be tested? How common is khat-chewing, in general, in Ethiopia? How common is it among women, compared to men? More context needs to be provided. Why is it a social activity? Why do people, particularly women, chew khat? Relatedly, how common is alcohol consumption among women in Ethiopia? Where is the Gambela region referenced in the Discussion section? And, your explanation as to why Gambela is different is unclear (English needs editing) – not sure what you’re trying to say. While fleshing out this section, include how the findings are relevant to more literature.

• Your conclusion states that rural residents were positively associated with HIV testing, while throughout the paper, you state it is urban residents. Review paper carefully for consistency (cleaning up the English will help with clarity). Can you offer concrete ideas for policymakers? How to target poor or uneducated women, etc.?

Reviewer #2: Thank you for the opportunity to review this manuscript. I believe the manuscript has the potential to contribute to the body of literature related to HIV testing in both rural Africa. I think the essence of your argument based on your findings is that people are understanding their risks and are going for testing. This however, is not clear in the arguments that you make. And so in terms of how you then phrase your argument, it would appear that going for testing is a bad thing which I don't believe is what you intend.

The discussion section is relatively short and void of the substance I was looking for in my review. For instance, what distinguishes urban dwellers from rural dwellers when it comes to HIV testing based on your findings? The conclusion was also not adequately developed.

6. PLOS authors have the option to publish the peer review history of their article (what does this mean?). If published, this will include your full peer review and any attached files.

Reviewer #1: No

Reviewer #2: No

---

## [Author Response · Author response to Decision Letter 0]

14 May 2020

Thank you very much for PLOS one editorial office, academic editors, as well as reviewers of this manuscript entitled with factors associated with premarital HIV testing among married women in Ethiopia for their astonished effort.

The written documents below explained point by point response for respective editors and reviewers

Academic editor’s comments and respective author response 

Editors comment 1. We noticed you have some minor occurrence of overlapping text with the following previous publication(s), which needs to be addressed:

https://www.unaids.org/en/regionscountries/countries/ethiopia

https://www.benthamopen.com/FULLTEXT/TOAIDJ-10-34

In your revision ensure you cite all your sources (including your own works), and quote or rephrase any duplicated text outside the methods section. Further consideration is dependent on these concerns being addressed.

Author response: based on the comments, the authors modified the overlapped text in the manuscript.

Editors comment 2. Please refrain from stating p values as 0.000, either report the exact value or employ the format p<0.001.

Author response: all p-value result in the manuscript which have a value of 0.000 were corrected as p<0.001.

Editors comment 3. In your Data Availability statement, you have not specified where the minimal data set underlying the results described in your manuscript can be found. PLOS defines a study's minimal data set as the underlying data used to reach the conclusions drawn in the manuscript and any additional data required to replicate the reported study findings in their entirety. All PLOS journals require that the minimal data set be made fully available. For more information about our data policy, please see http://journals.plos.org/plosone/s/data-availability.

Author response: For this analysis, we used the USAID–DHS program 2016 Ethiopian demographic and health survey data set. To request the same or different data for another purpose, a new research project request should be submitted to the DHS program here: https://dhsprogram.com/data/Access-Instructions.cfm. The DHS Program will normally review all data requests within 24 – 48 hours (during working days) and provide notification if access has been granted, or additional project information is needed before access can be granted. After receiving permission, the researcher can login and select the specific data in the format they prefer. 

Editors comment 4. PLOS requires an ORCID iD for the corresponding author in Editorial Manager on papers submitted after December 6th, 2016. Please ensure that you have an ORCID iD and that it is validated in Editorial Manager. To do this, go to ‘Update my Information’ (in the upper left-hand corner of the main menu), and click on the Fetch/Validate link next to the ORCID field. This will take you to the ORCID site and allow you to create a new iD or authenticate a pre-existing iD in Editorial Manager. Please see the following video for instructions on linking an ORCID iD to your Editorial Manager account: https://www.youtube.com/watch?v=_xcclfuvtxQ

Author response: the corresponding author linked his ORCID to editorial manager.

Editors comment 5. Your ethics statement must appear in the Methods section of your manuscript. If your ethics statement is written in any section besides the Methods, please move it to the Methods section and delete it from any other section. Please also ensure that your ethics statement is included in your manuscript, as the ethics section of your online submission will not be published alongside your manuscript.

 Author Response: the ethics statement is moved in the methods section of the manuscript based on the comments.

Additional Editor Comments (if provided):

Editors comment : Please ensure that the revised manuscript is sent for a full professional English proofread.

Author response: revised manuscript was sent for full professional English for proof read by English language expert.

Editor comment: Please include a completed STROBE checklist as part of the supplementary material for this article.

Author response: STROBE checklist as part of the supplementary material for this article was attached. 

Reviewer 1 comments and respective author response

This is an important topic and the analysis is solid. The manuscript, however, needs extensive editing for English. It is not publishable in its current state.

Reviewer comment 1: Include the URL of where the original data can be accessed as per PLOS ONE guidelines

Author response: For this analysis, we used the USAID–DHS program 2016 Ethiopian demographic and health survey data set. To request the same or different data for another purpose, a new research project request should be submitted to the DHS program here: https://dhsprogram.com/data/Access-Instructions.cfm. The DHS Program will normally review all data requests within 24 – 48 hours (during working days) and provide notification if access has been granted, or additional project information is needed before access can be granted. After receiving permission, the researcher can login and select the specific data in the format they prefer.

Reviewer comment 1: Add background information about premarital HIV testing. Is it required? Is it encouraged? If so, by whom, how and to what extent? In general, what percent of women undergo premarital HIV testing, and what percent of men (to compare)? Is premarital HIV testing an important imperative of the Ministry of Health?

Author response: Background information about premarital HIV testing is included in the revised manuscript. According to EDHS 2016 report, 24.5 % of married women age 15-49 ever tested before getting married or living with a partner but the men data about premarital HIV testing is not found for comparison. The ministry of health has accepted premarital voluntary HIV counseling and testing, which is recommended by World Health Organizations (WHO), since it is one of the key elements in the prevention and control of HIV/AIDS in the country and included in the revised manuscript.

Reviewer comment 2: Add information about gender-related factors in Ethiopia. While HIV prevalence is higher for women, what gender-related factors account for that? Are women more likely to be tested than men? Are women more at risk of contracting HIV due to gender-normative behavior?

Author response: information about gender-related factors affecting women to be at risk for HIV is included in the background. For instance, women are particularly vulnerable to HIV infection because of increased biologic susceptibility to HIV transmission through heterosexual sexual contact .Women are also at increased risk of HIV because they face a host of structural barriers and contextual gender inequalities such as poverty, economic disempowerment, cultural inequities, increased risk of sexual violence, and gender power imbalance in sexual interactions.

According to Ethiopia demographic and health Survey (EDHS) 2016 report, the proportion of women and men who were ever tested for HIV increased from 2% for women and men in 2005 to 20% for women and 21% for men in 2011. However, the HIV testing coverage remains unchanged between 2011 and 2016. The above information’s were narrated and included in the introduction section of the manuscript. 

Reviewer comment 3: The Discussion section should be fleshed out more. For example, testing rates are higher among urban residents (it appears that part of your paragraph is missing). Could this be because they have better/closer/easier access to testing facilities? Is stigma lower in urban areas leading more women to be tested? How common is khat-chewing, in general, in Ethiopia? How common is it among women, compared to men? More co ntext needs to be provided. Why is it a social activity? Why do people, particularly women, chew khat? Relatedly, how common is alcohol consumption among women in Ethiopia? Where is the Gambela region referenced in the Discussion section? And, your explanation as to why Gambela is different is unclear (English needs editing) – not sure what you’re trying to say. While fleshing out this section, include how the findings are relevant to more literature.

Author response: The discussion part is well narrated based on the comment which is provided for the authors. Considering residence, women who were residing in urban area have higher odds to undertake premarital HIV testing. The reason for this may be better availability and accessibility of HIV testing facilities in urban settings. Considering alcohol and khat chewing, according to EDHS 2016 report, the proportion of women who consumed alcohol and chewed chat in Ethiopia in the last 30 days was 50% and 65 %, respectively. As well, the present study revealed that premarital HIV testing was higher among khat chewer and alcohol drinker compared to their counterparts. This could be due to risky sexual behavior after alcohol and khat chewing. This may have increased perceived susceptibility to HIV which in turn leads them to be tested for HIV.

Regarding the Gambela region, it is one of 9 regions found in Ethiopia. The explanation for the discrepancy may be due to sample size variation, in which the current study was done based on nationally representative data. The study done in Gambela was region specific, which have small sample size. The above information is included more in the revised manuscript attached.

Reviewer comment 4: Your conclusion states that rural residents were positively associated with HIV testing, while throughout the paper, you state it is urban residents. Review paper carefully for consistency (cleaning up the English will help with clarity). Can you offer concrete ideas for policymakers? How to target poor or uneducated women, etc.?

Author response: the authors amended the conclusions based on the result consistently. 

Regarding poor or uneducated women, the Ethiopian government needs to step up efforts to expand education for all Women. 

Reviewer #2 comments and respective author response: 

Reviewer comment 1: Thank you for the opportunity to review this manuscript. I believe the manuscript has the potential to contribute to the body of literature related to HIV testing in both rural Africa. I think the essence of your argument based on your findings is that people understand their risks and are going for testing. This however, is not clear in the arguments that you make. And so in terms of how you then phrase your argument, it would appear that going for testing is a bad thing which I don't believe is what you intend.

Author response: Generally, premarital testing is the best solution for prevention, care, treatment, and support services. Make out and intervening thus factors enable for undertakes premarital HIV testing, which leads to trim down HIV acquisition among couples. Therefore, the endeavor of this study was to identify factors associated with premarital HIV testing among married women in Ethiopia. the above amended argument is included in the revised manuscript attached. 

Reviewer comment 2: The discussion section is relatively short and void of the substance I was looking for in my review. For instance, what distinguishes urban dwellers from rural dwellers when it comes to HIV testing based on your findings? The conclusion was also not adequately developed.

Author response: The discussion part is well narrated based on the comment which is provided for the authors. Considering residence, women who were residing in urban area have higher odds to undertake premarital HIV testing. The reason for this may be better availability and accessibility of HIV testing facilities in urban settings. The conclusion is adequately developed based on the comments and incorporated in the revised manuscript.

---

## [Decision Letter · Decision Letter 1]

8 Jun 2020

PONE-D-20-03022R1

Factors associated with premarital HIV testing among married women in Ethiopia.

PLOS ONE

Dear Dr. Ahmed,

Thank you for submitting your manuscript to PLOS ONE. After careful consideration, we feel that it has merit but does not fully meet PLOS ONE’s publication criteria as it currently stands. Therefore, we invite you to submit a revised version of the manuscript that addresses the points raised during the review process.

We look forward to receiving your revised manuscript.

Kind regards,

Benn Sartorius, PhD

Academic Editor

PLOS ONE

Reviewers' comments:

Reviewer's Responses to Questions

**Comments to the Author**

1. If the authors have adequately addressed your comments raised in a previous round of review and you feel that this manuscript is now acceptable for publication, you may indicate that here to bypass the “Comments to the Author” section, enter your conflict of interest statement in the “Confidential to Editor” section, and submit your "Accept" recommendation.

Reviewer #1: (No Response)

2. Is the manuscript technically sound, and do the data support the conclusions?

Reviewer #1: Yes

3. Has the statistical analysis been performed appropriately and rigorously? 

Reviewer #1: Yes

4. Have the authors made all data underlying the findings in their manuscript fully available?

Reviewer #1: No

5. Is the manuscript presented in an intelligible fashion and written in standard English?

Reviewer #1: No

6. Review Comments to the Author

Reviewer #1: As per PLOS ONE instructions, the authors need to "Describe where the data may be found in

full sentences." They provide this information in their response to the reviewer's comments, but it is not provided in the article submission. The manuscript continues to require editing for English.

7. PLOS authors have the option to publish the peer review history of their article (what does this mean?). If published, this will include your full peer review and any attached files.

Reviewer #1: No

---

## [Author Response · Author response to Decision Letter 1]

10 Jun 2020

Thank you very much for PLOS one editorial office, academic editors, as well as reviewers of this manuscript entitled with factors associated with premarital HIV testing among married women in Ethiopia for their astonished effort.

The written documents below explained point by point response for respective editors and reviewers

Reviewer 1 comments for authors: 

As per PLOS instructions the authors need to “describe where the data maybe found in full sentences” they provide this information in their response to the reviewer’s comments, but it is not provided in the article submission”. The manuscript continues to require to editing for English. 

Author response: For this analysis, we used the USAID–DHS program 2016 Ethiopian demographic and health survey data set. To request the same or different data for another purpose, a new research project request should be submitted to the DHS program here: https://dhsprogram.com/data/Access-Instructions.cfm. The DHS Program will normally review all data requests within 24 – 48 hours (during working days) and provide notification if access has been granted, or additional project information is needed before access can be granted. After receiving permission, the researcher can login and select the specific data in the format they prefer. The above information is included and attached in the revised manuscript.

The English language, grammar and spelling errors were corrected by linguist.

---

## [Editor Report · Decision Letter 2]

24 Jun 2020

Factors associated with premarital HIV testing among married women in Ethiopia.

PONE-D-20-03022R2

Dear Dr. Ahmed,

We’re pleased to inform you that your manuscript has been judged scientifically suitable for publication and will be formally accepted for publication once it meets all outstanding technical requirements.

Kind regards,

Benn Sartorius, PhD

Academic Editor

PLOS ONE
---

## [Editor Report · Acceptance letter]

30 Jun 2020

PONE-D-20-03022R2 

Factors associated with premarital HIV testing among married women in Ethiopia. 

Dear Dr. Ahmed:

I'm pleased to inform you that your manuscript has been deemed suitable for publication in PLOS ONE. Congratulations! Your manuscript is now with our production department. 

Kind regards, 

on behalf of

Dr. Benn Sartorius 

Academic Editor

PLOS ONE